

# Biocrust-linked changes in soil aggregate stability along a climatic gradient in the Chilean Coastal Range

Nicolás Riveras-Muñoz[1*], Steffen Seitz[1], Kristina Witzgall[2], Victoria Rodríguez[3], Peter Kühn[1],

Carsten W. Mueller[4], Rómulo Oses[5], Oscar Seguel[6], Dirk Wagner[3,7], and Thomas Scholten[1]

[1]University of Tübingen, Department of Geosciences, Soil Science and Geomorphology, Rümelinstr. 19–23, 72070 Tübingen, Germany
[2]Technical University of Munich, Soil Science, Emil-Ramann-Str. 2, 85354 Freising, Germany
[3]GFZ German Research Centre for Geosciences, Section 5.3 Geomicrobiology, Telegrafenberg, 14473 Potsdam, Germany
[4]Department of Geosciences and Natural Resource Management, University of Copenhagen, Copenhagen, Denmark
[5]Universidad de Atacama, Centro Regional de Investigación y Desarrollo Sustentable de Atacama (CRIDESAT), Copayapu 485, Copiapó, Chile
[6]Universidad de Chile, Facultad de Ciencias Agronómicas, Av. Santa Rosa #11315, La Pintana, 8820808 Santiago, Chile
[7]University of Potsdam, Institute of Geosciences, Karl-Liebknecht-Str. 24-25, 14476 Potsdam, Germany

*Correspondence to Nicolás Riveras-Muñoz* (nicolas-andres.riveras-munoz@uni-tuebingen.de)

**Abstract.** Biological soil crusts (biocrusts) composed of cyanobacteria, bacteria, algae, fungi, lichens, and bryophytes stabilize the soil surface. This effect has mainly been studied in arid climates, where biocrusts constitute the main biological agent to stabilize and connect soil aggregates. Besides, biocrusts are an integral part of the soil surface under mediterranean and humid climate conditions, mainly covering open spaces in forests and on denudated lands. They often develop after vegetation disturbances, when their ability to compete with vascular plants increases, acting as pioneer communities and affecting the stability of soil aggregates. To better understand how biocrusts mediate changes in soil aggregate stability under different climate conditions, we analyzed soil aggregate samples taken under biocrust communities from four national parks in Chile along a large climatic gradient ranging from (north to south) arid (Pan de Azúcar), semi-arid (Santa Gracia), mediterranean (La Campana) to humid (Nahuelbuta). Biocrust communities showed a stabilizing effect on the soil aggregates in dry fractions for the three northern and the wet aggregates for the southernmost sites. Here, permanent vascular plants and higher contents of organic carbon and nitrogen in the soil control aggregate stability more than biocrusts, which are in intense competition to higher plant communities. Moreover, we found an increase in stability for edge aggregate size classes (<2.0 mm and 9.5 – 30.0 mm). The geometric mean diameter of the soil aggregates showed a clear effect due to the climatic gradient, indicating that the aggregate stability presents a log-normal instead of a normal distribution, with a trend of low change between aggregate size fractions. Based on our results, we assume that biocrusts affect the soil structure in all climates. Their role for aggregate stability is masked under humid conditions by higher vegetation and organic matter contents in the topsoil.



## 1. Introduction

Biological soil crusts (biocrusts) are highly variable communities of microscopic (cyanobacteria, algae, fungi, and bacteria) and macroscopic (lichens, bryophytes) organisms found on the surface and in the upper centimeters of the soil (Gao et al., 2017). They stabilize the soil surface (Garcia-Pichel et al., 2016), especially in arid climates,

where biocrusts are the main biological agents for consolidating and connecting soil aggregates (Belnap and Büdel, 2016). However, biocrusts are also present in more mesic regions (e.g., pine barrens, serpentine soils, temperate steppe) (Belnap et al., 2016), but due to their limited ability to compete for light, they are mainly relegated to open spaces or interspaces between vascular plants where sunlight reaches the soil surface (Malam Issa et al., 1999).

Despite their simple structure, biocrusts are present in a wide variety of climatic conditions. Biocrust organisms lack specialized desiccation control structures, such as stomata or impermeable cuticles, so their water content depends on the humidity in the surrounding environment (Thielen et al., 2021). However, low water demand, high drought tolerance (Chen et al., 2020), the ability to grow actively only when water is available, and to recover without physiological damage even after complete drying out for an extended period (Oliver et al., 2005)

compensate the lack of dedicated structures. For this reason, biocrusts form an almost continuous layer in arid regions where water availability limits vascular plant cover (Colesie et al., 2014; Grote et al., 2010). By slightly increasing the water availability, areas covered by plants and biocrusts increase in self-organized patterns where both coexist. However, when water demand is not restrained anymore, vascular plants have an advantage in the use of light due to their canopy development, which leads to a decrease in biocrusts (Chen et al., 2018).

Biocrust cover and composition strongly depend on water availability (Bowker et al., 2016). Under dry conditions, with high potential evapotranspiration ($ET_0$), biocrusts are dominated by cyanobacteria, bacteria, and micro-fungi, with few bryophytes or lichens present. As $ET_0$ decreases, cyanobacterial biomass increases, and bryophytes and lichens appear. It implies that the external morphology of the biocrusts ranges from smooth to rugose (Chamizo et al., 2016). The water availability depends on the water-holding capacity of the soil, which depends on the parent

material and its degree of weathering and thus indirectly controls the distribution and composition of biocrusts at ecoregional and local scales (Bowker et al., 2016). For instance, Steven et al. (2013) showed that the composition of biocrust communities differed at vertical scales of a few centimeters in soils with different parent material origins, while Bowker et al. (2016) conclude that heterogeneous distributions in parent materials result in abrupt transitions in biocrust distribution and cover.

Biocrusts can be understood as an organic-sedimentary system within the topsoil where the inorganic and the organic part play dynamic roles in determining the architecture, evolution, and properties of the system, including structure and aggregate stability. On a small scale, biocrusts interact with the soil system in nitrogen and carbon cycling (Barger et al., 2016). The main processes of nitrogen enrichment are biological fixation and dust capture, while nitrogen losses typically appear via dissolution, volatilization, and erosional loss (Barger et al., 2016).

Photosynthesis is the most crucial carbon fixation process (Elbert et al., 2012; Porada et al., 2014), and soil erosion and biological decomposition are the primary loss source of carbon and other nutrients (Li et al., 2008).

Biocrusts affect soil erosion acting as a physical barrier that shields the soil from the direct exposition to water and wind (Seitz et al., 2016), providing protection to the effect of raindrops and thus splash erosion (Seitz et al.,



2017; Goebes et al., 2015) and modulating the abrasive effect of wind and surface runoff (Belnap and Büdel,

2016). At the same time, biocrusts control water flow across the landscape and through the soil matrix (Thielen et al., 2021; Eldridge et al., 2020). Eldridge et al. (2000) described a decrease in surface runoff and an increase in water infiltration in the presence of biocrusts under semi-arid conditions, related to a reduction in sediment discharge. The influence of biocrusts on the composition of the soil porosity is variable and depends on its stage of development and composition. In some cases, this structuration generates discontinuities that hinder the flow

of water in the soil, while in others, it generates a decrease in the tortuosity that is reflected in a rapid infiltration (Fischer et al., 2013). Water infiltration usually is inversely related to surface runoff (Lichner et al., 2012). The successional stage of biocrusts affects water repellency compared to bare soil (Drahorad et al., 2013). It has been observed that with the development of biocrusts, the water repellency increased, and the sorptivity and conductivity decreased (Fischer et al., 2012; Lichner et al., 2012). Therefore, biocrusts affect soil erosion and

hydrology through a wide variety of processes (Belnap and Büdel, 2016).

In regards to the stability of the soil surface, biocrusts further have a binding effect on aggregates and can form coherent structures (Belnap and Büdel, 2016). Typically, the organic carbon in the form of exo-polysaccharides or structural filaments of the different organisms present within biocrust communities causes soil stabilization (Garcia-Pichel et al., 2016). Other structure-forming processes involving biocrusts, although to a lesser extent, are

the compressive and drying action on the soil matrix and the pH-dependent dissolution of cementing salts (Bowker et al., 2016). The biocrusts-induced soil aggradation results in the formation of a defined layer, increasing the soil resistance and resilience to wind and water (Rosentreter et al., 2016).

Biocrusts stabilize individual aggregate units through different mechanisms depending on their species composition (Garcia-Pichel et al., 2016). For instance, bacteria play a crucial role in forming and stabilizing

aggregates by extracellular polymeric substances that glue soil particles together (Six et al., 2004). Vegetal debris serves as aggregation cores where the soil microorganisms use it as an energy source, but rapid decomposition is limited by the interaction with the inorganic matrix (Oades and Waters, 1991). On the other hand, fungi are important in forming soil aggregates due to their hyphal structure, which physically traps aggregates and soil particles (Totsche et al., 2018). Therefore, soil aggregate stabilization processes are dynamic and simultaneously

occur at different time and space scales (Oades and Waters, 1991; Tisdall and Oades, 1982).

However, soil aggregate stability depends not solely on biocrust growth and activity. Soil aggregation is determined and driven by a multitude of both biotic and abiotic factors. An increase in stability directly infers with the content of organic matter (Bartoli et al., 1992), clays (Scholten, 1997), calcium carbonates, gypsum, Fe- and Al-oxides, hydroxides, and oxyhydroxides (Dalal and Bridge, 2020; Oades and Waters, 1991). The

electrolytes present in the soil solution have a variable effect on aggregation stability, mainly depending on ion valence and ion specificity, expressed in characteristics such as electrical conductivity, type of cations, sodium adsorption ratio, pH, etc. (Liu et al., 2020). Also, external soil factors can modify the stability of aggregates. For example, the climate in the short term controls the cycles of wetting and drying (semi-arid and sub-humid regions), freezing, and thawing (temperate regions). Nevertheless, limited detailed and quantitative information is available

on the association between the composition of the humic substances and aggregate distribution for soils spanning a broad range of climatic gradients (i.e., temperature and precipitation). Few studies address the stability of soil aggregates along a climatic gradient directly. For example, Jing et al. (2021) studied the stability of aggregates



along a precipitation gradient (semi-arid: 390 mm y$^{-1}$ to semi-humid: 526 mm y$^{-1}$) along the Loess Plateau in China, where the semi-humid condition showed an increase in the stability of aggregates. Wei et al. (2020)

investigated aggregation along a climatic gradient (temperate: MAT 13.7°C, MAP: 650 mm y$^{-1}$; tropical: MAT 24.2°C, MAP: 1722 mm y$^{-1}$ with heavy texture soils (silty clay loam, silty clay, and clay) under constant land use in central-southern China and found an increase in macroaggregates from the temperate region (MAT: 13.7°C, MAP 650 mm y-1) and the subtropical (MAT:16.8°C, MAP: 1577 mm y$^{-1}$) and then a steady decrease until the tropical region (MAT°C: 24.2°C, MAP: 1722 mm y$^{-1}$). The connection between aggregate stability, biota, and

organic matter dynamics mostly depends on climate (Amézketa, 1999; Six et al., 2004). In the long term, climate affects the structural stability of the soil via the composition and dynamics of vegetation (Amézketa, 1999). Moreover, pedogenesis and the age of the soil have a cumulative effect on soil structuring given by processes like accumulation of organic matter, clay migration, and acidification (Lin, 2011). Finally, land use and soil management determine the structure and stability of the soil surface by magnitude, frequency, and duration of

external disturbances, for example, plowing (Bronick and Lal, 2005).

In this study, we investigate how and to what extent biocrusts under different climatic conditions stabilize the soil surface. Therefore, we compare the stability of macroaggregates and varying soil properties in topsoil with or without biocrust cover along a climatic gradient from arid to humid climate conditions along the Chilean Coastal Range. We test the following hypotheses:

(i) if biocrusts cover the soil surface, soil aggregates show a higher stability because the biocrusts protect the soil surface physically, shelter soil organic matter within aggregates, modify the structure of microbial communities, and change water flow into the soil,

(ii) if the climate is arid, the effect of biocrusts on the soil surface is most pronounced because other sources of organic matter are at minimum and biocrusts represent the main soil cover, and

(iii) if the humidity of the climate increases, the stabilizing effects of biocrust on the soil surface are reduced, although without disappearing entirely, because water availability increases and higher vegetation hinder the growth of biocrusts.



## 2. Materials and methods

### 2.1. Study sites and experimental settings

In order to assess the climatic effect on soil and its interactions with biocrusts, four study sites distributed between latitudes from 26°06′S to 37°48′S and over 1300 km were established in the Chilean Coastal Range: Pan de Azúcar National Park (PA), Santa Gracia Natural Reserve (SG), La Campana National Park (LC) and Nahuelbuta National Park (NA), corresponding to arid, semi-arid, mediterranean and humid climates, respectively (Bernhard et al., 2018).

The study sites are comparable in geology, geomorphology, land use, and glacial and volcanic influence (Bernhard et al., 2018). The parent material in all the study sites is granitoid, keeping this factor of soil formation nearly constant along the studied gradient (Oeser et al., 2018). The dominant topography is generally fluvial valleys, and the sites had no glacial influence during the last glaciation (Hulton et al., 2002). The sites are located within nature protection areas, with limited anthropogenic influence compared to the surrounding areas. Despite this, the
occasional entry of cows to LC (Rundel and Weisser, 1975) and goats to SG (Armesto et al., 2007) has been reported. These conditions allow us to focus on the environmental effect on two other soil-forming factors, i.e., climate and vegetation.

The mean annual temperature (MAT) decreases from north to south (PA: 16.8 °C, SG: 13.7 °C, LC: 14.1 °C, NA: 6.6 °C). The mean annual precipitation (MAP) in the study sites increases from north to south (PA: 12 mm yr$^{-1}$,
SG: 66 mm yr$^{-1}$, LC: 367 mm yr$^{-1}$, NA: 1469 mm yr$^{-1}$) with similar rainfalls distribution mostly distributed in winter months (May to August) (Bernhard et al., 2018). The elevation of the sites increases from north to south (PA: 329 – 351 m a.s.l., SG: 642 – 720 m a.s.l., LC: 708 – 732 m a.s.l., NA: 1200 – 1270 m a.s.l.). Paleoclimate modeling studies (Mutz et al., 2018) indicate that these climate patterns have been persistent since the late Pliocene; thus, the study sites represent the long-term impact of climate on the soil (Ewing et al., 2006). Bernhard
et al. (2018) classified soils in the study sites as Regosols in PA, Cambisols for SG and LC, and Umbrisols in NA. In general, pedogenic processes such as soil depth, clay contents, organic matter accumulation, porosity, and activity ratio are correlated with the humidity of the site.

For each of the 4 study sites, 5 plots of 1 x 1 m were established as replicates. Each plot was located in the top-slope position with south-facing exposition, considering the presence of site-typical biological soil crust
communities, similar slope and aspect, a lack of anthropogenic disturbance, and a maximum distance of 30 m between each plot. Each plot included patches with at least the size of the samples with 100% biocrust cover (BSC+), and additionally, a nearby point without biocrust cover (BSC-) was defined as control.

### 2.2. Biocrust sampling and classification

Biocrust patches of approximately 100 cm$^2$ were collected in the field and stored in paper envelopes before
classification for every research plot. Sampled communities showed all typical biocrust classes from cyanobacteria, algae, fungi, lichens, liverworts, and mosses. The species composition further showed a graduating change from lichen-dominating biocrusts in the northernmost site to bryophyte-dominating biocrusts in the southernmost site. Dominant macroscopic biocrust species were determined for each of the four sites to the genus level by morphological characteristics using a stereomicroscope (Leitz TS, Wetzlar, Germany), a transmitted-light





microscope (Leitz Laborlux S, Wetzlar, Germany), and ultraviolet light. Species groups were separated into bryophytes (Lightowlers, 1985; He, 1998; Ochyra and Matteri, 2001; Cuvertino et al., 2012; Fariña and Ardiles, 2014) and lichens (Galloway and Quilhot, 2009) and assigned to the different regions (Table 1). A more detailed survey and classification of individual species, including algae and cyanobacteria, will be sought for further studies.

**Table 1.** Taxonomical composition of mosses and lichens in the biological soil crust for the study sites along the climatic gradient.

| Site / Division | Family | Genus | Number of species |
|---|---|---|---|
| **PA** | | | |
| **_Lichens_** | Cladoniaceae | _Cladonia sp._ | 2 |
| | Verrucariaceae | _Placidium sp._ | 2 |
| | Lecanoraceae | _Lecidella sp._ | 1 |
| | Rhizocarpaceae | _Rhizocarpon sp._ | 1 |
| **SG** | | | |
| **_Mosses_** | Pottiaceae | _Syntrichia sp._ | 2 |
| | Pottiaceae | _Tortella sp._ | 2 |
| Lichen genera unidentified | | | 2 |
| **LC** | | | |
| **_Mosses_** | Bartramiaceae | _Philonotis sp._ | 1 |
| | Bryaceae | _Bryum sp._ | 1 |
| | Pottiaceae | _Syntrichia sp._ | 2 |
| | Pottiaceae | _Tortella sp._ | 2 |
| Moss and lichen genera unidentified | | | 2 + 1 |
| **NA** | | | |
| **_Mosses_** | Amblystegiaceae | _Acrocladium sp._ | 1 |
| | Amblystegiaceae | _Amblystegium sp._ | 1 |
| | Bartramiaceae | _Bartramia sp._ | 1 |
| | Bryaceae | _Bryum sp._ | 1 |
| | Dicranaceae | _Campylopus sp._ | 2 |
| | Pterigynandraceae | _Myurella sp._ | 1 |
| Liverwort, lichen and fungi genera unidentified | | | 2 + 2 + 1 |

### 2.3. Soil sampling and analyses

Bulk topsoil samples (0 – 5 cm) were taken with metal-core sample augers directly under biocrust patches and in comparative zones without biocrust cover and sieved to 2 mm after air drying.

Bulk density (BD) and soil water content were determined gravimetrically. The particle size distribution was determined for seven fractions according to (1929) combining sieving of fractions >20 μm and pipetting of fractions <20 μm. Soil texture was interpreted according to the WRB soil classification system (Jahn et al., 2006).



Soil pH was determined in water by a WTW pH 340 (WTW GmbH, Weilheim, Germany) using a Sentix 81 electrode, and electrical conductivity was measured with a conductivity meter (LE703, Mettler Toledo, USA). Total carbon ($C_t$) and nitrogen ($N_t$) were analyzed using oxidative heat combustion at 1150 °C in a Vario EL III elemental analyzer (Elementar Analysensysteme GmbH, Hanau, Germany).


The physical stability of soil aggregates was measured to quantify the destructive effect of water and mechanical forces through two-stage sieving: dry and wet (Hartge and Horn, 2009). The water-stable aggregates were measured by sieving undisturbed air-dried soil samples through a pile of sieves of decreasing mesh size (19.0,

14.7, 9.8, 6.8, 4.8, 3.3, and 2.0 mm) then, by repeating the same process underwater. Finally, the values for each size were calculated relative to the initial sample and corrected by coarse fragments content. With this aggregate stability data, accumulated frequency curves were calculated, and a set of indexes were estimated: mean weight diameter of the aggregates (ΔMWD) (Hartge and Horn, 2009; Van Bavel, 1950), Geometric mean diameter (ΔGMD) (Mazurak, 1950), water stability aggregate ratio (WSAR) (Liu et al., 2014) and the proportion of soil

macroaggregates of a diameter less than 2 mm ($R_{<2\ mm}$) (Liang et al., 2015) as described below.

*Difference in mean weight diameter (ΔMWD):*

$$\Delta MWD = \left.\sum_{i=1}^{n} X_i * W_i \middle/ \sum_{i=1}^{n} W_i\right.$$

(1)

where $W_i$ is the corrected mass proportion of stable aggregate fraction $i$ in the total 2–30 mm aggregate and $X_i$ is the mean diameter of stable aggregate fraction i.

*Geometric mean diameter (ΔGMD):*

$$\Delta GMD = \exp\left[\left(\sum_{i=1}^{n} W_i \lg X_i\right)\middle/\left(\sum_{i=1}^{n} W_i\right)\right]$$

(2)

where $X_i$ is the sieve opening size (mm); $W_i$ is the proportion of the total sample mass occurring in the $i$-size fraction; $n$ is the number of particle fractions.

*Water Stability Aggregate Ratio:*

$$WSAR(\%) = WSA/A * 100$$                                (3)

where $WSA$ is the >2 mm water-stable aggregate content and $A$ is the >2 mm dry aggregate content.

*Proportion of soil macroaggregate of a diameter less than 2 mm*

$$R_{<2\ mm} = \frac{M_{r>2}}{M_T} * 100 = \left(1 - \frac{M_{r<2}}{M_T}\right)$$

(4)

where $M_{r<2}$ is the weight of macroaggregates with a diameter less than 2 mm, $M_T$ is the total sample weight, $M_{r<2}$

is the weight of microaggregates with a diameter less than 2 mm.



### 2.4. Statistical analyses

The influence of the climatic gradient (study site) and biocrust presence on physicochemical soil parameters and aggregate stability in 40 plots (4 study sites, 2 biocrust treatment, 5 replicates) were assessed by factorial generalized linear models (GLM) because of the lack of normal distribution for most of the variables according to the Shapiro-Wilk test. The link functions used for each model were selected based on the lowest Akaike information criterion (AIC) selection and characteristics of the data (skewness, counts, continuous variables, proportions) between Gaussian, Gamma, inverse Gaussian, and Tweedie distributions. Differences in treatments were tested using pair-wise F-tests with $p < 0.001$ as significance criteria. The analyses where conducted in *R* 4.2.0 (Team, 2018) and the GLM distributions were extended from the base R core with the *Tweedie* 2.3.3 package (Dunn, 2017). All visualizations were made with the package *ggplot2* 3.3.3 (Wickham, 2016).



## 3. Results

### 3.1. Soil properties

Soil pH was significantly affected by the climatic gradient (Table 2), with mean values of 7.7 in PA, 6.2 in SG, 5.9 in LC, and 4.4 in NA, with acidification levels of 6.2 in BSC- to 5.9 in BSC+. In terms of electrical conductivity (EC), a remarkably higher value of 2646.1 µS cm$^{-1}$ in PA is outstanding in comparison with the low and homogeneous values of 109.3 µS cm$^{-1}$ for SG, 153.8 µS cm$^{-1}$ for LC, and 102.3 µS cm$^{-1}$ for NA (Table 2). EC did not differ for the biocrust treatment. Bulk density (BD) showed a significant difference between the study sites (Table 2), with higher values in the two dryer sites, with 1.5 g cm$^{-3}$ in PA, and 1.6 g cm$^{-3}$ in SG, and a decrease in the more humid sites, with 1.2 g cm$^{-3}$ in LC and 0.6 g cm$^{-3}$ in NA (Table 2).

Total nitrogen ($N_t$) content was directly proportional to the climatic gradient, with values of 0.04% for PA, 0.07% for SG, 0.28% for LC and 0.51% for NA (Table 2). A slightly similar pattern was observed for the total carbon ($C_t$) (Table 2), with values of 1.1% in PA, 0.8% in SG, 5.0% in LC and 12.5% in NA, but with a significant decrease when the BSC+ is present, from 5.6% to 4.2% in average. The relation between $C_t$ and $N_t$, expressed as C/N ratio, showed significantly different values of 33.9 in PA, 12.3 in SG, 16.7 in LC, and 24.5 in NA on average; and with a significant decrease from 27.0 in BSC- to 16.7 in BSC+ (Table 2). It is important to note that although PA and NA present the highest values, the condition changes diametrically when observed together with the BSC+ treatment, with a large dispersion in PA and stable values in NA.

The distribution of the soil particle size classes did not show clear patterns along the climatic gradient, with PA deviating from it in all cases. Despite this, the observed values were significant, with clay values of 9.6% for PA, 7.3% in SG, 10.4% in LC and 24.6% in NA; silt with 28.9% in PA, 18.7% in SG, 20.0% in LC and 21.9% in NA; and sand with 61.5% in PA, 73.9% in SG, 69.6% in LC and 53.5% in NA. When biocrusts were present, a significant decrease from 13.6% to 12.6% in clays and an increase from 21.2% to 23.6% in silt could be observed with higher dispersion for the arid site (PA).



250

**Table 2.** Soil properties without biocrust cover (BSC-) and with (BSC+) for Pan de Azúcar (PA), Santa Gracia (SG), La Campana (LC) and Nahuelbuta (NA) (EC = electrical conductivity, BD = bulk density, $N_t$ = total nitrogen, $C_t$ = total carbon, C/N = (carbon to nitrogen ratio).

| Site | Biocrust | pH | EC (µS cm⁻¹) | BD (g cm⁻³) | $N_t$ (%) | $C_t$ (%) | C/N | Clay (%) | Silt (%) | Sand (%) |
|------|----------|-----|----|----|----|----|----|----|----|----|
| PA | BSC- | 8.0 ± 0.1 | 2952.5 ± 1413.9 | 1.7 ± 0.1 | 0.03 ± 0.01 | 1.46 ± 0.52 | 53.6 ± 16.2 | 13.6 ± 2.5 | 25.8 ± 5.8 | 60.6 ± 5.4 |
|  | BSC+ | 7.4 ± 0.3 | 2339.6 ± 888.9 | 1.3 ± 0.3 | 0.06 ± 0.01 | 0.80 ± 0.48 | 14.1 ± 8.2 | 5.6 ± 2.2 | 32.1 ± 6.2 | 62.3 ± 4.3 |
| SG | BSC- | 6.2 ± 0.1 | 111.0 ± 46.6 | 1.6 ± 0.1 | 0.07 ± 0.03 | 0.84 ± 0.33 | 12.4 ± 1.2 | 7.2 ± 1.7 | 18.2 ± 3.0 | 74.6 ± 4.4 |
|  | BSC+ | 6.2 ± 0.1 | 107.6 ± 26.7 | 1.6 ± 0.1 | 0.06 ± 0.02 | 0.79 ± 0.25 | 12.2 ± 1.1 | 7.5 ± 1.2 | 19.2 ± 2.4 | 73.3 ± 3.5 |
| LC | BSC- | 5.9 ± 0.7 | 158.1 ± 95.9 | 1.1 ± 0.2 | 0.32 ± 0.15 | 6.14 ± 4.82 | 17.7 ± 4.2 | 10.1 ± 1.0 | 19.9 ± 4.1 | 70.0 ± 4.9 |
|  | BSC+ | 5.8 ± 0.2 | 149.5 ± 32.7 | 1.2 ± 0.2 | 0.24 ± 0.06 | 3.91 ± 1.64 | 15.7 ± 3.5 | 10.8 ± 1.3 | 20.0 ± 1.2 | 69.2 ± 2.4 |
| NA | BSC- | 4.5 ± 0.4 | 102.5 ± 31.4 | 0.6 ± 0.1 | 0.57 ± 0.10 | 13.91 ± 3.13 | 24.2 ± 2.6 | 23.7 ± 1.1 | 20.9 ± 2.1 | 55.4 ± 3.0 |
|  | BSC+ | 4.2 ± 0.3 | 102.2 ± 5.4 | 0.6 ± 0.1 | 0.45 ± 0.08 | 11.12 ± 2.53 | 24.8 ± 2.3 | 25.4 ± 1.3 | 22.9 ± 4.2 | 51.7 ± 5.2 |



### 3.2. Soil aggregate stability

To determine the stability of the soil macroaggregates along the climatic gradient together with the biocrust influence, dry sieving showed a significant difference between study sites, but not for the biocrust treatment (Table 3). Dry aggregates in the 19.0 – 30.0 mm range showed significantly different values of 4.4% in PA, 1.2% in SG, 5.5% in LC, and 36.2% in NA. The fraction 9.5 – 19.0 mm revealed differences in the interaction between study sites and biocrusts, increasing the proportion of aggregates from 13.2% in PA to 19.3% SG and 29.3% in LC, not in NA with only 2.9% in the presence of biocrusts. The aggregates between 6.7 – 9.5 mm showed a significant decrease from 13.3% in PA, 5.2% in SG, 6.5% in LC, and 4.1% in NA. In interaction with biocrusts, it showed a significant increase in the proportion when it was present. The fraction from 4.7 – 6.7mm showed significantly different values of 13.6% in PA, 3.4 % in SG, 5.0% in LC, and 5.6% in NA, while the interaction with biocrusts showed a significant increase when is present (BSC+). Aggregates from 3.4 – 4.7 mm size showed significance among study sites, with 9.8% in PA, 3.6% in SG, 4.4% in LC, and 7.9% in NA. Aggregates between 2.0 – 3.4 mm showed a slightly similar amount of 9.9% in PA, 5.9% in LC, and 14.3% in NA, but with a minor proportion of 4.9% in LC, while when biocrust is present in LC, it showed a slight decrease. Finally, for the dry sieved aggregates under 2 mm, there was a significant reduction for the study sites, with values of 69.1% in SG and 47.3% in LC relative to 30.1% in PA, but not in NA with 28.9%; while the biocrust effect in interaction with the site is significant with 60.6% in SG and 40.8% in LC, indicating the same proportion of aggregates in this site for SG with 30.0% and NA with 39.8%.

In a second stage, aggregate stability under wet conditions was characterized, with a clear difference between sites, while biocrusts had a significant effect only on the edge aggregate size classes (Table 3). At the same time, the fraction 19.0 – 30.0 mm showed an increasing significant pattern in the amount of aggregates, with 2.8% in NA, 0.8% in SG, 8.9% in LC and 29.9% in NA, while biocrusts significantly increased the proportion of aggregates by 35.6%, from 9.0% for BSC- to 12.2% for BSC+. For the fraction 9.5 – 19.0 mm, there were significant differences between the study sites, the biocrust effect, and its interactions, with a notable increase from 5.4% to 11.7% on average when biocrusts were present. The wet sieved aggregates in the range of 6.7 – 9.5 mm showed significant differences only between the study sites, with 10.6% in PA, 2.5% in SG, 4.2% in LC and 4.3% in NA. Wet aggregates in the range of 4.7 – 6.7 mm showed significant differences between the study sites with 10.6% in PA, 2.3% in SG, 3.7% in LC and 5.8 in NA. The fraction between 3.4 – 4.7 mm showed only significant differences between the study sites, with 6.9% in PA, 2.2% in SG, 3.0% in LC, and 6.1% in NA. Aggregates ranging from 2.0 – 3.4 mm showed differences between the study sites, with 5.8% in PA, 3.2% in SG, 4.2% in LC, and 6.8% in NA. The fractions under 2.0 mm were significantly different for the study sites with 58.3% in PA, 79.6% in SG, 62.4% in LC, and 41.3% in NA; and for biocrusts from 63.9% for BSC+ treatment to 56.9% for BSC- treatment.



285 **Table 3.** Aggregate proportion for dry and wet sieved fractions for Pan de Azúcar (PA), Santa Gracia (SG), La Campana (LC) and Nahuelbuta (NA) for biocrust (BSC+) and non-biocrust (BSC-) treatments.

| Site | Biocrust | 19.0 – 30.0 mm | 9.5 – 19.0 mm | 6.7 – 9.5 mm | 4.7 – 6.7 mm | 3.4 – 4.7 mm | 2.0 – 3.4 mm | >2.0 mm |
|---|---|---|---|---|---|---|---|---|
| | | **Dry aggregates (%)** | | | | | | |
| PA | BSC- | 2.8 ± 2.8 | 24.3 ± 8.1 | 16.5 ± 3.3 | 16.6 ± 1.6 | 10.1 ± 1.6 | 8.9 ± 0.5 | 20.5 ± 6.1 |
| | BSC+ | 6.0 ± 10.2 | 13.2 ± 8.3 | 10.2 ± 3.0 | 10.5 ± 2.9 | 9.4 ± 2.1 | 10.9 ± 1.8 | 39.8 ± 14.5 |
| SG | BSC- | 0.0 ± 0.0 | 6.0 ± 4.6 | 4.3 ± 2.2 | 3.1 ± 0.9 | 3.8 ± 1.3 | 5.2 ± 1.7 | 77.6 ± 5.6 |
| | BSC+ | 2.3 ± 3.2 | 19.3 ± 7.6 | 6.0 ± 1.1 | 3.7 ± 0.7 | 3.3 ± 0.6 | 4.6 ± 0.6 | 60.6 ± 8.1 |
| LC | BSC- | 4.9 ± 4.9 | 21.5 ± 20.3 | 5.0 ± 3.2 | 4.3 ± 2.5 | 4.2 ± 1.1 | 6.1 ± 2.0 | 53.8 ± 27.7 |
| | BSC+ | 6.0 ± 9.6 | 29.3 ± 16.5 | 7.9 ± 4.1 | 5.6 ± 1.9 | 4.7 ± 1.4 | 5.7 ± 1.6 | 40.8 ± 24.4 |
| NA | BSC- | 35.9 ± 12.4 | 5.7 ± 2.4 | 5.0 ± 0.5 | 6.5 ± 2.4 | 8.1 ± 3.4 | 11.6 ± 4.5 | 27.8 ± 6.8 |
| | BSC+ | 36.5 ± 21.4 | 2.9 ± 2.6 | 3.2 ± 2.3 | 4.7 ± 1.7 | 7.7 ± 3.2 | 16.9 ± 6.2 | 30.0 ± 13.8 |
| | | **Wet aggregates (%)** | | | | | | |
| PA | BSC- | 0.0 ± 0.0 | 0.9 ± 1.3 | 14.0 ± 2.6 | 12.3 ± 3.3 | 7.1 ± 2.3 | 6.4 ± 3.0 | 59.4 ± 7.5 |
| | BSC+ | 5.6 ± 9.1 | 9.1 ± 6.9 | 7.2 ± 4.8 | 8.9 ± 2.1 | 6.7 ± 1.4 | 5.2 ± 2.4 | 57.2 ± 6.0 |
| SG | BSC- | 0.1 ± 0.1 | 3.4 ± 3.2 | 1.9 ± 1.6 | 2.1 ± 1.4 | 2.6 ± 1.7 | 3.4 ± 1.3 | 86.5 ± 6.8 |
| | BSC+ | 1.6 ± 2.5 | 15.3 ± 6.0 | 3.2 ± 1.6 | 2.4 ± 1.0 | 1.9 ± 0.6 | 2.9 ± 0.6 | 72.8 ± 3.8 |
| LC | BSC- | 5.9 ± 4.3 | 10.6 ± 9.2 | 3.7 ± 2.7 | 3.7 ± 1.7 | 3.3 ± 1.3 | 4.2 ± 1.0 | 69.8 ± 17.6 |
| | BSC+ | 11.8 ± 12.2 | 17.7 ± 10.6 | 4.7 ± 2.1 | 3.7 ± 1.1 | 2.8 ± 0.7 | 4.2 ± 1.5 | 55.1 ± 20.4 |
| NA | BSC- | 30.1 ± 12.4 | 6.7 ± 4.7 | 4.7 ± 1.4 | 6.2 ± 3.2 | 6.0 ± 3.0 | 6.3 ± 2.7 | 40.1 ± 5.6 |
| | BSC+ | 29.8 ± 13.1 | 4.7 ± 5.4 | 3.9 ± 3.8 | 5.5 ± 3.0 | 6.2 ± 3.7 | 7.4 ± 4.2 | 42.6 ± 12.6 |





When comparing the changes of the aggregate distributions between wet and dry conditions (Figure 1), an irregular pattern was observed, with a general decrease in most of the analyzed fractions, except for an increase
290    in the amount of aggregates of 19.0 mm for NA. This was even higher than for the BSC+ treatment. It is important to mention that NA also showed a slight increase in the proportion of 9.5 and 6.7 mm aggregates.



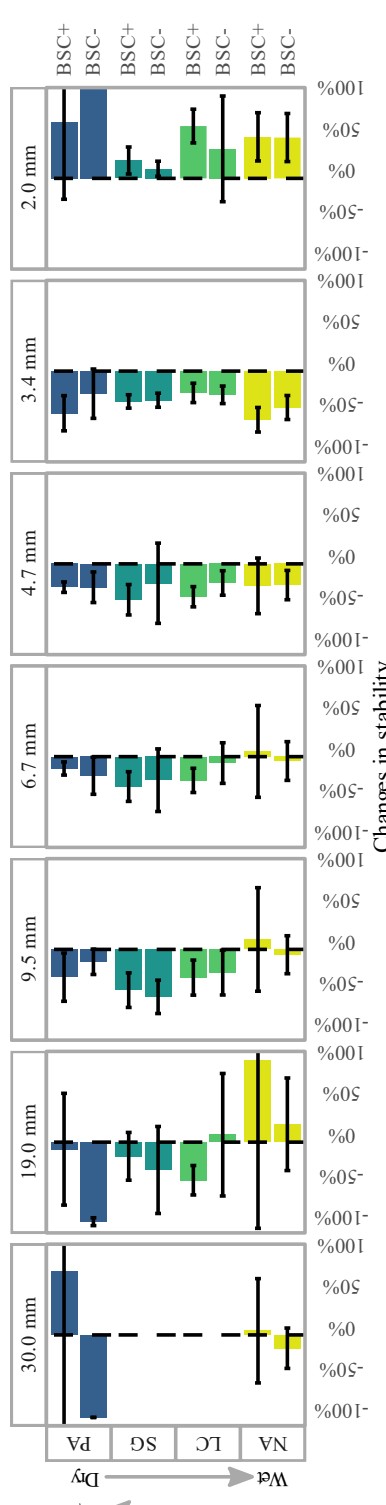

**Figure 1.** Aggregate size distribution changes between dry and wet sieving for the sites Pan de Azúcar (PA), Santa Gracia (SG), La Campana (LC), and Nahuelbuta (NA) for biocrust (BSC+) and non-biocrust (BSC-) treatments. Error bars were, in some cases, truncated for better visualization.



Soil aggregate stability was evaluated through different indexes to integrate the different sizes and sieving conditions in a summary value. The difference in mean weight diameter of the aggregates ($\Delta$MWD) showed no significance in any of the conditions (Table 4). However, there was a significant difference in the variation of the geometric mean diameter of the aggregates ($\Delta$GMD) for the study sites, with values of 1.86 mm for PA, 1.2 mm for SG, 1.4 mm for LC, in comparison with the more stable condition of 0.83 mm for NA. The water-stable aggregates ratio was significant for the study sites, showing differences between NA with 81.1% and the other study sites (57.7% PA, 66.2% SG, and 73.4% LC). The ratio of soil macroaggregates of a diameter less than 2.0 mm ($R_{<2\ mm}$) presents differences in the study sites and for biocrust treatments. SG showed a value of $R_{<2\ mm}$ of 79.6% and NA of 41.9%, which was different from PA and LC with 58.6% and 62.4%, respectively. For biocrust treatments, $R_{<2\ mm}$ changed from 63.7% to 57.5% with biocrusts, indicating a biocrust-induced decrease in the proportion for this fraction. Finally, according to the analyzed indexes, NA showed the most stable conditions, and alternating PA and SG showed the most unstable conditions.





**Table 4.** Aggregate stability indexes for Pan de Azúcar (PA), Santa Gracia (SG), La Campana (LC) and Nahuelbuta (NA) for biocrust (BSC+) and non-biocrust (BSC-) treatments. $\Delta$MWD = mean weight diameter, $\Delta$GMD = geometric mean diameter, WSAR = water-stable aggregate ratio, $R_{<2\,mm}$ = ratio of aggregates less than 2mm.

| Site | Biocrust | $\Delta$MWD (mm) | $\Delta$GMD (mm) | WSAR (%) | $R_{<2\,mm}$ (%) |
|------|----------|------------------|------------------|----------|------------------|
| PA | BSC- | 4.9 ± 1.6 | 2.4 ± 0.5 | 41.6 ± 24.5 | 59.4 ± 7.5 |
|    | BSC+ | 1.1 ± 1.6 | 1.3 ± 0.6 | 73.8 ± 17.0 | 57.2 ± 6.0 |
| SG | BSC- | 0.6 ± 0.7 | 1.2 ± 0.2 | 61.6 ± 27.0 | 86.5 ± 6.8 |
|    | BSC+ | 1.0 ± 1.1 | 1.3 ± 0.2 | 70.8 ± 13.1 | 72.8 ± 3.8 |
| LC | BSC- | 1.7 ± 2.2 | 1.6 ± 0.7 | 71.3 ± 20.1 | 69.8 ± 17.6 |
|    | BSC+ | 0.6 ± 0.9 | 1.2 ± 0.2 | 75.4 ± 7.0 | 55.1 ± 20.4 |
| NA | BSC- | 1.4 ± 1.6 | 0.8 ± 0.1 | 83.2 ± 5.8 | 39.0 ± 5.7 |
|    | BSC+ | 1.6 ± 2.2 | 0.8 ± 0.1 | 79.0 ± 9.8 | 44.9 ± 13.3 |



As shown above, the climatic gradient (site) had significant effects for clay, silt, sand, pH, electrical conductivity (EC), bulk density (BD), total carbon ($C_t$) and nitrogen ($N_t$), C/N ratio, dry and wet aggregates under 30 mm, $\Delta$GMD, WSAR, $R_{<2\,mm}$. Biocrust treatments were significant for clay, silt, pH, total carbon ($C_t$) and nitrogen ($N_t$), C/N ratio, and wet aggregates from 9.5 to 30.0 mm and >2 mm, and $R_{<2\,mm}$. Finally, the interaction of the site and biocrusts was significant for clay, bulk density (BD), total nitrogen ($N_t$), dry aggregates from 4.7 to 19.0 mm and 0 to 3.4 mm, wet aggregates from 9.5 to 30.0 mm, and $\Delta$MWD.



## 4. Discussion

Biocrusts are spread along a wide variety of climate conditions, being the dominant soil cover under arid
conditions and relegated to areas between vascular plants or to disturbed areas in mesic and humid conditions
(Malam Issa et al., 1999). The main reason for that unequal distribution is competition: In arid conditions,
biocrusts dominate due to their reduced water demand compared to vascular plants (Belnap et al., 2016). However,
plants quickly cover the soil when its water demand is supplied, hindering the development of biocrusts due to
the lower amount of light reaching the soil surface (Chen et al., 2018).

Many benefits have been attributed to the biocrusts presence, i.e., nitrogen fixation, hydrological modifications in
infiltration and runoff, erosion reduction (Chamizo et al., 2016), soil aggregation, and structuration (Garcia-Pichel
et al., 2016), etc. Nevertheless, due to the dominant presence in arid conditions, limited knowledge exists about if
those functions significantly persist under humid and mesic conditions (Seitz et al., 2017; Corbin and Thiet, 2020).

### 4.1. Aggregate stability and soil properties along the climatic gradient

The climatic gradient has a significant effect on the stability of soil aggregates. Using the geometric mean diameter
($\Delta$GMD), an index that replaces the linear fitting of $\Delta$MWD with a logarithmic one, significant differences for the
study sites along the climatic gradient can be observed ($p$-value: <0.001). Interestingly, if soil aggregate stability
was evaluated with the difference in mean weight diameter ($\Delta$MWD), it did not show significant changes along
the climatic gradient. WSAR, an index that shows the ratio of aggregates that persist stable after disruption by
water, showed a similar behavior as $\Delta$MWD. The main difference between $\Delta$MWD and $\Delta$GMD is that $\Delta$GMD
performs better in non-uniform particulate substances (Hatch and Choate, 1929), which corresponds to soils
equilibrated in the content of sand, silt, and clay (Mazurak, 1950) and pointing soil texture indirectly as a critical
factor in aggregate stability along the climatic gradient. Further, considering the $\Delta$GMD data, an increase in
stability was observed as moving along the climatic gradient to a higher water availability condition, but with SG
going out of that pattern with a more stable condition. In PA, in the drier north, the condition proved to be less
stable than NA, which was statistically in a different category with the most stable condition. SG presented the
highest ratio of unstable aggregates under the studied range of sizes (highest $R_{<2mm}$) and NA the lowest, with close
to half of it, indicating augmented aggregate stability in the complementary range of sizes.

The effect of the climatic gradient is not only expressed in the stability of soil aggregates, but it is also present
with different intensities in a variety of soil properties. The pH decreases continuously from the northern arid to
the southern humid study site in accordance with Bernhard et al. (2018). The high pH in PA can be attributed to
the constant input of atmospheric aerosols, e.g., salts, gypsum, and calcium carbonates (Ewing et al., 2006) in
combination with the arid climate that allows salts to accumulate in the topsoil (Slessarev et al., 2016). Whereas
in the southern sites, the forwarding increase of precipitations results in a reduction in the pH due to leaching of
soluble salts (Slessarev et al., 2016) and an increase in soil respiration (Orchard and Cook, 1983). The
accumulation of soluble salts is well established for the arid site PA, as saline conditions (Allison and Richards,
1954) are indicated by the high electrical conductivity value. These higher amounts of salts have a strong effect
in structure degradation dynamics, linked to the destabilizing effect of sodium and stabilizing of carbonates
(Corwin, 2021). Although $C_t$ and $N_t$ follow the climatic gradient, when comparing the C/N ratio, PA and NA have
higher values. High values of the C/N ratio indicate a nitrogen limitation of plants and other organisms (Brust,





2019), pointing out that this occurs at the two opposite sites along the climatic gradient. This could be explained by the biological activity (Zhang et al., 2013), which may be close to a physiological limitation in the case of NA, while for PA, it may indeed be due to low nitrogen availability (Hooper and Johnson, 1999). However, Bernhard et al. (2018) also indicate a high amount of carbonates in PA, property not measured in this study, which makes

PA hardly comparable in terms of $C_t$. Despite the properties following the climatic gradient, SG deviates from the other sites in terms of higher bulk density (BD), lower clay, and higher sand content. This can indicate a degraded condition for the semi-arid site caused by the current land use, including grazing (Armesto et al., 2007), compacting the surface and thus activating erosive processes (Scholten and Seitz, 2019), in favor of the accumulation of sand particles (Govers, 1985). The aggregate distribution stresses this finding, where SG has a

lower proportion of water-stable aggregates >2 mm and a higher water-destabilization of aggregates between 9.5 to 4.7 mm, indicating the nature of the structuring agent in that zone is water-soluble.

Further, soil properties did not fully follow the climatic gradient, agreeing with Bernhard et al. (2018), who described varying results in pedogenic and microbial processes. They were significant along the gradient for clay content, surface horizon and soil profile depth, total organic carbon, DNA amounts, BD, pH, and base saturation.

This lack of significance indicates that further factors are not included in the study design and affect those characteristics. The soil formation factors: time, parent material, and topography (Dokuchaev, 1883) were set constant by the study design, allowing to study the influence of organisms and climate variables on the system. Nevertheless, applying this partitioning of the model of soil formation factors to a natural system may result in the addition of noise. For our experiment, the climate was included as the climatic gradient and organisms as the

biocrusts effect; however, the latter only represents a small part of the possible effects of biota. In this sense, it is interesting to evaluate the effect of biocrusts under controlled environment conditions in the laboratory.

### 4.2.  Biocrusts altering soil properties along the climatic gradient

Despite these factors beyond the climatic gradient, biocrusts showed effects on clay, silt, pH, total carbon ($C_t$), total nitrogen ($N_t$), C/N ratio, and wet aggregates from 9.5 to 30.0 mm and >2 mm. However, as this was an

observational study, it only allows establishing associations between factors and not cause-effect relationships (Cox, 1992; Rosenbaum, 2005). It is thus possible that changes in soil characteristics promote the biocrust establishment, as well as that biocrust establishment triggers changes in these properties (Belnap and Lange, 2003).

The biocrust treatments showed a significant decrease in pH (*p*-value: 0.002404), reflecting the biological activity of its constituent organisms, which acidifies the soil due to the carbon dioxide released by cellular respiration

(Bachar et al., 2010). The pH values reported by Bernhard et al. (2018) are in the same range as ours but without differentiating between BSC+ and BSC-, as this factor was not part of their study. The content of $C_t$ and $N_t$ were significantly different when biocrusts were present, but it did not affect any of the aggregate sizes or stability indexes. In this sense, biocrusts play a role in the carbon and nitrogen cycles (Chen et al., 2000), as they are formed mainly by photosynthetic and nitrogen-fixing organisms (Maestre et al., 2013), but it has not an immediate impact

on the soil aggregate stability and points that the main stabilizing agent is of organic origin (Wagner et al., 2007; Six et al., 2004).

Considering the stabilizing effect of biocrusts on wet sieved aggregates between 9.5 and 30.0 mm, we could show that it occurs prominently at the three northern sites, whereas in NA there was no difference with and without



biocrusts. This points to a threshold in the biocrust-induced stabilization of the soil aggregates between LC and
NA and partially confirms our initial assumption that biocrusts have the greatest effect in arid conditions. However, the effect on aggregate stability for the wet condition varies according to the variable used, being specific for limited aggregate sizes in terms of mechanical disturbances (dry sieving), but without a substantial improvement concerning water stability (wet sieving). This lack of difference in wet sieved aggregate point to a non-soluble nature of the stabilizing agents, which can be attributed to stabilization due to organic structures and exudates
(Rillig, 2004), and stress the idea that NA differs to the other sites in the mechanisms of aggregate stabilization as a local adaptation, where due to the higher proportion of precipitation, is conducted by water stable mechanisms.

Although soil aggregate stability showed differences along the climatic gradient through ΔGMD, when considering the effect of biocrusts, it only shows differences for $R_{<2mm}$, an indicator of values outside the study range. This lack of significance suggests either a smaller than expected or the methodology used fails to detect
such an effect of biocrusts on soil aggregate stability along the climatic gradient. However, the proportion of aggregates outside the study range ($R_{<2mm}$) and the difference on wet sieved aggregates between 9.5 and 30.0 mm by biocrusts give clues on biocrust-induced stabilization of soil aggregates. Nevertheless, these changes in soil stability due to biocrusts may not be noticeable because they are outside the size range measured or because the effect is being masked in the summary indices used due to a minor role of the size fractions where it occurs.

The results indicate that soil aggregate stabilization mechanisms are different in PA than at the other sites. With that in mind, it was found that in PA, biocrusts grow in areas with a lower content of clay and higher content of silt, which implies increased nutrient availability and water holding capacity (Chen et al., 2000), while the sand fraction was not related. However, the method used can amplify that difference since the determination of particle size distribution does not consider coarse fragments (Köhn, 1929), which were abundant at PA. In addition, the
soil covered with biocrusts showed a lower value for bulk density (BD) only for PA, while in the other sites, this property was not affected. This can be interpreted as a biocrust-induced decrease in soil density due to increased intra- and extra-aggregate porosity and organic matter (Whitney et al., 2017) or that biocrusts grow under the least limiting condition (Bowker et al., 2014). Soils with biocrust cover showed a trend of lower electrical conductivity, which can be explained by inhibition of biocrusts by toxicity due to the accumulation of salts in the soil, or to the
consumption of salts as a source of nutrients by the organisms in the biocrusts (Abed et al., 2019).

Biocrust plays a role along the climatic gradient affecting different properties, *i.e.,* clay, silt, pH, total carbon ($C_t$), total nitrogen ($N_t$), C/N ratio, and wet aggregates from 9.5 to 30.0 mm and >2 mm. Nevertheless, the way that each property change responds to local conditions: In the arid northernmost site, there is a strong influence of the salts in terms of stabilization and establishment of biocrusts, while at the southernmost site, there is no stabilization
of the aggregates, but a contribution to the carbon and nitrogen contents. The most consistent property along the climatic gradient was pH, indicator of biological activity. However, with biological activity on this scale, no difference is made between plants, microorganisms, fungi, bacteria, etc. Finally, considering the largest size of the persistent wet aggregates match with the characteristics attributed to fungi and bryophytes, capable of retaining micro-aggregates and soil particles between their hyphae (Kleber et al., 2007; Six et al., 2004; Totsche et al.,
2018) and points to this as the most significant mechanism of soil aggregate stabilization of biocrusts.





## 5. Conclusions

This study aimed to investigate how and to what extent biocrusts stabilize soil aggregates along a climatic gradient (arid to humid) in Chile, based on the assumption that biocrusts induce aggregation on the soil under arid conditions, where they act as the main stabilizing agents, and thus can express their effects more clearly.

Biocrusts increased stability for wet aggregates and edge size classes ranging from arid to mediterranean climates and improved aggregation in wet and dry conditions for most aggregates ranging between 2.0 and 30.0 mm. We can thus partly confirm our first hypothesis (i) with the restriction that not all aggregate classes were affected in all conditions, and therefore no clear effect could be described. Furthermore, it became clear that the stabilizing effect of climate on soil aggregates was more pronounced than the effect of biocrusts.

The biocrust effect on aggregates stability was less pronounced under humid climate conditions, attributable to a stabilization effect by water-stable substances and variable composition of biocrust communities, which can differ in the stabilization mechanisms. This indicates a transition in the main biotic agents driving the aggregation and its mechanisms on the soil surface, moving from biocrust communities in arid conditions to vascular plants in humid conditions and confirms our second (ii) and third (iii) hypotheses. According to the metric used, the biocrust

effect changes after disturbance by sieving, being specific for limited aggregate sizes in terms of dry sieving stability but to a minor degree for water sieving stability.

When looking at complementary soil properties along a climatic gradient that includes arid, semi-arid, mediterranean, and humid climates, these showed a wide variety of trends and patterns not always correlated with climate, indicating more factors affecting such processes. The climatic gradient affected most soil properties

studied, such as pH, electrical conductivity (EC), bulk density (BD), $C_t$, $N_t$, C/N ratio, clay, silt, and sand content. Nevertheless, when analyzing the biocrust effect, the differences were limited to a few properties like $C_t$, $N_t$, C/N ratio, clay, sand, which could indicate that the accumulation processes of organic matter occur at a faster rate than aggregation processes. The biocrust effect was more substantial for most of the properties under arid conditions, decreasing as the water availability increased, and thus our second hypothesis can be partly confirmed (ii).

Moreover, it became clear that local and particular soil conditions result in varying soil and biocrust characteristics. These are, for example, high EC from salt in PA, signs of physical degradation due to grazing in SG, and different structuring mechanisms in NA due to the diametrically different water regime and high OM content.

Finally, we can conclude the biocrusts in our study area to be a valuable agent in stabilizing the upper topsoil layer, but for a narrow spectrum of conditions and mostly under arid conditions. Therefore, the effect could be

considered a transitory situation in ecological succession towards a stable ecosystem. In this process, biocrusts improve conditions for other more demanding species such as vascular plants, initially quickly improving the availability of carbon and nitrogen in the soil.





**Appendices**

Appendix A. Significant factors for response variables based on generalized linear models. Models with significant interaction also include the predictors as simple parameters based on marginality principle. ($p$-value = 0: "***"; $p$-value = 0.001: "**"; $p$-value = 0.01: "*").

| Dependent variable | Distribution (link-function for GLM) | Significance for independent variable | | |
|---|---|---|---|---|
| | | Site | Biocrust | Site × Biocrust |
| Clay | Gaussian | *** | * | |
| Silt | Inverse Gaussian | *** | * | |
| Sand | Gaussian | *** | | |
| Fine Silt | Tweedie | *** | | * |
| Medium silt | Tweedie | *** | *** | * |
| Coarse silt | Tweedie | *** | | |
| Very fine sand | Gaussian | *** | | |
| Fine sand | Tweedie | *** | | |
| Medium sand | Gaussian | *** | | |
| Coarse sand | Tweedie | *** | | |
| pH | Gaussian | *** | ** | |
| EC | Tweedie | *** | | |
| BD | Gaussian | *** | | ** |
| $N_t$ | Gamma | *** | | ** |
| $C_t$ | Tweedie | *** | ** | |
| C/N | Tweedie | *** | ** | |
| 19.0-30.0 mm Dry | Tweedie | *** | | |
| 9.5-19.0 mm Dry | Tweedie | *** | | ** |
| 6.7-9.5 mm Dry | Tweedie | *** | | ** |
| 4.7-6.7 mm Dry | Tweedie | *** | | * |
| 3.4-4.7 mm Dry | Tweedie | *** | | |
| 2.0-3.4 mm Dry | Tweedie | *** | | * |
| > 2.0 mm Dry | Tweedie | *** | | *** |
| 19.0-30.0 mm Wet | Tweedie | *** | * | ** |
| 9.5-19.0 mm Wet | Tweedie | * | ** | * |
| 6.7-9.5 mm Wet | Tweedie | *** | | |
| 4.7-6.7 mm Wet | Tweedie | *** | | |
| 3.4-4.7 mm Wet | Tweedie | *** | | |
| 2.0-3.4 mm Wet | Tweedie | *** | | |
| > 2.0 mm Wet | Tweedie | *** | * | |
| ΔMWD | Gaussian | | | |
| ΔGMD | Tweedie | *** | | |
| WSAR | Gaussian | | | |
| $R_{<2\ mm}$ | Gaussian | *** | * | |



**Data availability**

The data that support the findings of this study are available from the corresponding author upon request.

**Code availability**

The code that supports the findings of this study are available from the corresponding author upon request.

**Author contribution**

TS, DW, CWM, OS, and StS conceptualized the study. OS, KW, and NRM collected the soil samples. StS collected and analyzed the biocrust samples, and NRM analyzed the soil samples. NRM NRM performed the analyses and prepared the manuscript.

**Competing interests**

The authors declare that they have no conflict of interest.

**Acknowledgments**

We are deeply grateful to the students, technical assistants, and colleagues who helped in the field and laboratory phases. A special thanks to Martin Nebel and Sonja Thielen for their help in classifying bryophytes and lichens. We are grateful to the Chilean National Forest Corporation (CONAF) and Sucesión Gálvez Muñoz, who provided access to the study sites. The study was funded by the German Science Foundation DFGSPP 1803 (EarthShape; 485 www.earthshape.net).|





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
