# Peer review of "Biocrust-linked changes in soil aggregate stability along a climatic gradient in the Chilean Coastal Range"

_SOIL, 2021_

## Author Response (AR1)

Thank you very much for your answer to the reviewer comments. I was not very satisfied with the answer you gave to reviewer 2 concerning the role of carbon types for the phenomenon you want to study. For aggregation is quite a difference of organic or inorganic C are present and both carbon types should therefore be determined. I hope that you can address this point in your discussion. Also, you will need to adequately address all other reviewer comments.

> *Dear editor,*
>
> *We appreciate the helpful feedback and your support during the submission process.*
>
> *Since we still had samples saved, we decided to analyze the inorganic carbon, the results and their discussion have been included in the paper. For details, please see below our answers to reviewer 1 and 2.*
>
> *Best regards,*
>
> *Nicolás Riveras-Muñoz, on behalf of all co-authors*

Comments from reviewer 1

The study by Riveras-Muñoz et al. describes dry and wet aggregate stability along a climatic gradient in Chile and evaluates the effect of biocrusts on this stability. Dealing with these two factors, climatic gradient and biocrusts, the paper is multidisciplinary and should be of broad international interest.

However, I have some comments that I would like the authors to consider when revising their manuscript. These are:

**General comments**

Given that in the present study aggregate stability was analyzed and discussed with regard to (i) the effect of climate and further (ii) the biological influence, all sections including the introduction and abstract as well as hypotheses should not start with BSC but should follow this order: aggregate stability in general, effects of climate, effects of BSC.

Also, a strict order within sentences or paragraphs will improve readability of the text. This applies for the order of factors (climate/BSC), treatments (BSC-, BSC+) and sites (north to south).

Some sentences are very complicated in structure and content. Hence, I suggest to split these sentences for better reading and understanding.

> *Dear reviewer, we appreciate the time spent and the very enriching feedback to improve the document. We are going to apply the proposed structure and to follow the suggested sequence. In some parts of the discussion, it is not fully applicable because the factors climate and BSC and topography are interrelated in several sections of the text. However, the suggested order was incorporated as a reference and large parts of the manuscript will be restructured. We will also shorten long sentences (a*

*Chilean/German specialty :-) for better reading and understanding with the help of a native speaker.*

*We also thank the reviewer for the very helpful and detailed specific comments. We are going to change the text accordingly and give answers to your questions.*

**Specific comments**

Line 31 and 271: what are "edge aggregate size classes"? please explain

*It refers to the size of aggregates at the upper and lower limits of the classes used. We changed the text for better understanding.*

Line 55 ff: please, consider the fact that the occurrence of lichens is not restricted to more humid locations but that many lichens were also found in PA (e.g., Jung et al. 2019: DOI: 10.1111/gbi.12368; Jung et al., 2020: doi.org/10.1016/j.isci.2020.101647)

*Wording has been changed and the suggested citation has been added to better include lichens for arid locations.*

Line 59: please consider: available water for lichen growth can also mainly be provided by fog and dew (e.g., Jung et al, 2019: DOI: 10.1002/mbo3.894)

*Thank you very much for this comment, we added the information to the introduction section.*

Line 67: please explain: what do you mean by "small scale"? Elbert et al. 2012 (DOI: 10.1038/NGEO1486) pointed out that globally cryptogamic covers take up 3.9 Pg C per year.

*Here, "small" refers to spatial dimension. We clarified the text in this respect. However, we thank you for this comment and add the citation you gave to show the enormous importance of biocrust in global C cycles.*

Line 89: do you mean "soil structure-forming processes DUE TO biocrusts"?

*Yes, we do. Your suggestion has been incorporated in the text.*

Line 94: not only cyanobacteria but also green-algae are well known to produce extrapolymeric substances which can glue soil particles together (e.g. Lewin 1956: doi.org/10.1139/m56-079); please add

*Thank you for the suggested literature. We added it and the sentence was written in a clearer way.*

Line 98: what do you mean by "which physically traps aggregates and soil particles"? Please explain

*We refer to the fact that it physically traps due to its entangled structure to differentiate it from other forms of binding. The sentence has been rewritten to improve readability.*

Line 99: please clarify: because of what are soil aggregate stabilization processes dynamic and occur on different time and space scales?

*We added the explanation phrase to improve readability. "..., where an aggregate of soil particles is built up of structural units of various sizes held together by various binding agents."*

Line 107: what do you mean by "external soil factors", please explain

*Thank you for the comment. We change the text to "Also climatic processes can modify the stability of aggregates, for example cycles of wetting and drying in semi-arid and sub-humid regions or freezing and thawing in temperate regions".*

Line 108: why only "in the short term"? please explain

*We agree that "in the short term" is misleading and rephrased the sentence (see above comment).*

line 133: what do you mean by "on the soil surface"? please specify

*Here, we mean that biocrusts cover the soil surface which is a typical way to describe soils with biocrusts in soil science. Since we understand that this might be misleading to readers other than soil scientists and/or others that might understand biocrusts as part of the soil and, thus, the soil surface in this respect is the surface that either consists of a topsoil with or without a biocrust, we just deleted "on the soil surface".*

line 136: do not confine any of this hypothesis but discuss later in the discussion

*Well, this is a very difficult comment. In an earlier version of the manuscript we kept the hypothesis open as you suggest. The editor in charge of our manuscript decided that we should rephrase the hypothesis to the if-then-because form, before it could be send to reviewers. We did so and are now afraid to say that we should keep them as is following previous suggestion.*

line 169: it does not become clear how biocrusts were collected exactly – please explain;

*Thank you for the comment. We changed the text accordingly and added more details on sampling in chap. 2.2.*

please show pictures of the biocrusts sampled at each site (plot picture)

*Pictures of the sampled biocrust at each site have been included (see figure 1).*

line 177: for the time being: please refer to other researcher's work on algae and cyanobacteria in BSCs of these locations (e.g., Baumann et al., 2018: doi.org/10.1016/j.soilbio.2018.09.035; Samolov et al., 2020: doi:10.3390/microorganisms8071047)

*Dear reviewer, Thank you very much for the suggested literature, we included the most relevant points for our study as biocrust cover and algae and cyanobacteria species richness based on those publications.*

line 183: please describe exactly how and up to which depth biocrusts were removed (e.g., were the rhizoids of the mosses cut at the soil surface?); what did the BSC look like in PA and what exactly was sampled?

*Suggestions have been included giving more details of the methodology (see comment above, and beginning of chap. 2.2). Bryophytes were sampled with rhizoids down to 5 mm depth; all other species down to a depth of 2 mm. Please also see newly added figure 1.*

Line 190: Total C and N of what were analyzed? Please describe more precisely

*We changed the text to "Total carbon (Ct) and nitrogen (Nt) of the bulk topsoil samples (0 – 5 cm) were analyzed using oxidative heat combustion at 1150 °C in a Vario EL III elemental analyzer (Elementar Analysensysteme GmbH, Hanau, Germany)."*

Line 195ff: please explain: what do you mean by "…and corrected by coarse fragments content"?

*The term "coarse fragment" refers to any particle with a diameter greater than 2 mm and smaller than 250 mm (boulders). The whole sentence has been rephrased to "Finally, the values for each aggregate size were calculated relative to the initial sample weight after removing coarse fragments."*

Line 197 f: insert "difference" before "mean" and "geometric"; please spent some more sentences to explain what results of these indices will mean

*Thank you for this comment. We inserted "difference" and explained the meaning of the indices. The section now reads:*

*"…and a set of indexes were estimated: difference in mean weight diameter of the aggregates (ΔMWD) (Hartge and Horn, 2009; Van Bavel, 1950), difference in geometric mean diameter (ΔGMD) (Mazurak, 1950), water stability aggregate ratio (WSAR) (Liu et al., 2014) and the proportion of soil macroaggregates of a diameter less than 2 mm (R<2 mm) (Liang et al., 2015) as described below. ΔMWD and ΔGMD indicate how much the average diameter of soil aggregates changes between dry and wet conditions. The main difference between ΔMWD and ΔGMD is that the first considers a linear behavior between the different aggregate size classes, while the ΔGMD considers a logarithmic behavior.".*

Line 204: insert "fraction" after "2-30 mm aggregate"

*The suggestion has been included.*

Line 205: insert "differences" before "geometric"

*The suggestion has been included.*

Line 206: explain what "exp" means

*"exp" corresponds to the natural anti-logarithm operation, i.e. "e to the power of…"*

Line 209 and 212: insert abbreviation in brackets

*The suggestion has been included.*

Line 214f: is "weight" correct or should it be "mass"?

*Adjustments have been included. It has been expressed in different ways to respect the original source, but all indices used are based on weight.*

Line 253f: start paragraph with "Dry sieving showed….."

*The suggestion has been included.*

Tables: please insert in the captions that significant factors for response variables are given in the appendix

*The suggestion has been included.*

Fig 1: improve Fig. by noting BSC+ and BSC- also on left hand site of Fig.

*The suggestion has been included.*

Line 319-328: These paragraphs are repetitive and should be removed from here

*Thank you for your comment. We removed these two paragraphs.*

Line 340: what is does mean: "with a more stable condition"? please explain

*We rephrased the sentence to "The lower value of ΔGMD for SG indicates a comparably higher aggregate stability as it would be expected when we assume a steady trend from arid to humid climate".*

Line 341: which "different category"? please explain

*Thank you for this comment. We deleted "different category" since it is misleading in this context.*

Line 370 f: what do you mean by this sentence? Please reword

*When reading this paragraph, we felt the same as you and after some discussion among the authors, we decided the remove this paragraph. Originally, we wanted to point to noise in our data, that is related to factors that are not covered by the study design. However, since we didn't measure them, it is very vague to argue in this direction. Future publications from our ongoing laboratory experiments with microorganisms and biocrust might clarify some of these aspects.*

Line 375 f: "In this sense…..": the deduction of this idea is not clear, please explain

*See above.*

Line 403: what do you mean by "an indicator of values outside the study range"?, please explain

*The aggregates analyzed are in the range of 2 to 30 mm. However, it is possible that those aggregates break down into smaller diameters than 2 mm, so-called microaggregates. In this study, we did not separate this size class further into finer units. Hence, it is at the rim of the size class distribution that we applied. We rephrased to paragraph to "Soil aggregate stability showed clear differences along the climatic gradient. However, when considering the effect of biocrusts, differences were limited to the smallest aggregate size class ($R_{<2mm}$) referring to changes in microaggregate size distribution as described by Totsche et al. (2018) Further, difference for wet sieved aggregates with and without biocrusts between 9.5 and 30.0 mm point to biocrust-induced stabilization of soil aggregates."*

Line 404 f: split this sentence into two

*The suggestion has been included to improve readability.*

Line 416 f: for this statement it is very important to know how biocrusts looked like and were sampled and what exactly was included for bulk density, especially for samples in PA; please see grit porosity as outlined in Jung et al. 2019 and 2020

*This is a very interesting comment, as it refers to the newly described grit-sized biocenosis recently discovered by Jung et al. These grit-crusts were described on an escarpment directly on the coast in PA and, according to our understanding, are formed in the direct influence of the coastal fog around small stones and pebbles. Our sampling took place much further inland, around the actual main working area of the EarthShape consortium. As far as we can tell, we did not find and sample any grit-crust type BSCs here, so we described a patchily distributed classic biological or cryptogamic soil crust dominated by cyanobacteria. Please also refer to Figure 1. However, as this comment on the grit-crust concept covers a cutting-edge research topic in which we are admittedly not experts, we are happy to discuss further with the reviewer.*

Line 426: what do you mean by "no differences is made between...."? Please explain

*This is indeed unclear. We rephrased the sentence to "However, at the scale of the climatic gradient it is not possible to distinguish the origin of biological activity between plants, micro-organisms, fungi, bacteria, etc." to improve readability.*

Line 429: bryophytes have no hyphae but rhizoids – please add

*The suggestion has been included.*

Line 437: please repeat text of hypothesis

*The suggestion has been included.*

Line 442 ff: please split sentence

*The suggestion has been included to improve readability.*

Line 444 ff: please reword sentence and repeat hypotheses

*The suggestion has been included to improve readability.*

Line 449: what other factors do you mean? Please speculate

*The suggestion has been included to improve readability.*

Line 454: please repeat text of hypothesis

*The suggestion has been included to improve readability.*

Line 455: please reword the sentence to show its importance

*The suggestion has been included to improve readability.*

**Technical corrections**

Line 116: close bracket before "with"

*The suggestion has been included.*

Line 186: please insert name of reference

*The suggestion has been included.*

Line 198: use lower case g for "geometric"

*The suggestion has been included.*

Line 359: insert "a" before "property"

*The suggestion has been included.*

Line 378 f and 421 f: do not spell out total carbon and nitrogen again but use abbreviations

*The suggestion has been included.*

Line 390: insert "out" after "points"

*The suggestion has been included.*

Line 452: replace "," by "and" before "sand"

*The suggestion has been included.*

Comments from reviewer 2

The importance of biocrust on soil aggregate stability is underexplored. The research covers a large climate gradient transect of ecological relevance.

There are some issues with the paper which warrant comments or improvements from the full author team.

*Dear reviewer,*

*We were delighted to see that our manuscript is an interesting topic for you, and we thank you very much for your valuable suggestions for improving our manuscript. We appreciate the time you took to review this preprint.*

*Following, we respond to all comments and explain the specific changes made in detail.*

*Best regards,*

*Nicolás Riveras-Muñoz on behalf of all co-authors*

**Specific comments**

There are some issues with the paper which warrant comments or improvements from the full author team. The fact that only Ct is measured is somewhat of mistake. It is clear that in the southern site NA most of C will be organic C and in PA iit be inorganic C (carbonate) which will have differing effect on the aggregate structure, stability and formation.

*Regarding the measurement of total carbon, we agree that the origin of the carbon along the climatic gradient is important for the discussion of our results on the role of biocrusts for soil aggregate stability. Since inorganic and organic carbon have been measured for all sites within the EarthShape project (see Bernhard et al. 2018 for details,* https://doi.org/10.1016/j.catena.2018.06.018) *we now calculated the organic fraction for the northernmost study site as well. The description of the methodology was adopted (line 216 ff) and figure 2 now includes visualization for the results of soil organic and inorganic carbon (SOC and SIC). In the results section (line 269 to 272), SIC and SOC contents are described. Changes have been made along the discussion to incorporate the high carbonate content in PA.*

The other issue is whether or not we are observing the effect of pure lichen biocrust (PA) versus a moss containing or dominating biocrust (SG, LC and NA) on aggregates.

*This is a very interesting question and definitely an issue we, or other groups working on feedbacks between biocrusts and soil or ecosystems, should follow in future. One could imagine a set up that allows to clearly distinguish between different taxa in a lab or greenhouse experiment. However, we used biocrust patches sampled directly from their natural environment, aiming on the functional aspect of strengthening aggregate's stability in order the prevent soil from being eroded. Therefore, aggregate stability was measured as an expression of the resistance to physical forces acting on the aggregates in the course of dry and wet sieving, sometimes called dispersive energy. This was done on samples taken along a climatic gradient rather than comparing pure lichen biocrust (PA) versus a moss-containing or dominating biocrust (SG, LC, and NA) or taxonomical composition of mosses and lichens in the biological soil crust for the study sites as shown in Table 2. We ask for your understanding that we presented the taxonomic information in our paper, although it has not been used directly in our analysis, since we think that it is valuable, at least for those readers that come from the field of biological soil crust research.*

It is not 100% clear from the paper what the relevance is of the selected aggregate size, it feels somewhat random. Are all these macro-aggregates if so this should be stated. It is concerning that the aggregate size which show increased stability are the largest and smallest

size bins selected, a leftover effect? Also, to some extend visible in Fig 1 where 30 mm and 2.0 mm size fractions look complete different from all others, even without considering the biocrust effect.

> *Thank you for this comment, we are happy to explain why we have chosen the aggregate size classes as given in our paper and how we interpret the two edge size classes. At least in soil science (for details see Six et al. 2000, https://doi.org/10.2136/sssaj2000.642681x), macroaggregates, and this applies to all considered size classes in our paper, are typically divided in size classes with a decreasing class width to cover different environmental effect and functions. Macroaggregates are used for analyses if erosion, including wind and water erosion, is a factor to be assessed, since their mass decides whether the material is eroded or not. Another aspect which is of importance for our study is that the proportion of fresh particulate organic matter is much higher in macroaggregates compared to microaggregates. Since we hypothesize that soil aggregates from sites with biocrust cover are more stable than those from bare soil and organic matter is one of the main stabilizing components of topsoil, fresh intact or slightly decomposed organic matter from biocrust incorporated in macroaggregates is of high relevance for our study. Our description of the wet and dry sieving method and related sieve sizes was maybe a little too short since from our perspective it is a well-established method in soil physics. We have expanded the description to address the objection of leftover effects. Before dry and wet sieving, the field samples are dry sieved to leave out aggregates with sizes above 30.0 mm and below 2.0 mm. Following, the two edge size classes ranging from 30,0 to 19.0 and from 3.4 to 2.0 mm do not contain leftovers from aggregates sized above or below the typical size of macroaggregates. However, it is an important observation that you made and we added some explanatory text about the aggregate size class distribution of our samples as displayed in figure 3. The primary particles are the starting point for the aggregate size distribution in the soil. Depending on size, shape and roughness, different sized aggregates are formed. The organic matter, reinforced by the action of plant roots, mycorrhizae, and earthworms, acts as a glue between the aggregates composed of primary particles. The higher the proportion of organic matter, the larger the aggregates (see Nahelbuta in Figure 3). The larger the proportion of well-rounded grains, predominantly in the sand fraction, the smaller the primary aggregates and also those that are joined together by the organic matter. This effect is nicely reflected in the aggregates size class distribution (figure 3), with a sand content in SG and LC being significantly higher than in NA and PA (figure 2).*

The paper decribes in minute detail difference between fractions and sometimes looses the bigger picture. The visualisation of the data is minimal, most data is tabulated. With only one Figure. In all tables and the figure it is not clear what differences are actually significant in the four sites between treatment (biocrust) and for the various size fractions. Errors on the means (standard deviation or standard error?) are generally larger making it even more difficult to see what is of real significance and what is not. The lack of visualisation make the paper unappealing for the reader to look at.

> *Thank you for the comment. We have revised the text accordingly and tried to make the details less prominent. Tables have been replaced for figures, including mean, standard error/standard deviation, and letter-based display for significant differences.*

*Table 2 has been replaced by Figure 2*

*Table 3 has been replaced by Figure 3*

*Table 4 has been replaced by Figure 5*

The conclusion reads extremely long a full page. Most is (semi)-discussion and should go there.

> *We agree. Redundant and discussion-like statements have been removed to shorten the conclusion. Nevertheless, we kept some introductory paragraphs that are not part of it, to improve the reading of the conclusion independent of the rest of the text. For details, see lines 453 ff.*

I think it good to go back to some of the original work relevant to this study but this leads to reference of papers from 1950, 1929 and ultimately 1883. This is not fully balanced by lots of recent papers on the topic last 5 years.

> *References that are too old have been complemented with more up-to-date literature on the same subjects. To make clear that valuable and still valid research has aleady been done much earlier that the 21th century we pointed more clearly to the two excellent review papers from Johan Six and his research group added as follows:*
>
> *Six, J., Paustian, K., Elliott, E. T., and Combrink, C.: Soil Structure and Organic Matter I. Distribution of Aggregate-Size Classes and Aggregate-Associated Carbon, Soil Science Society of America Journal, 64, 681-689, https://doi.org/10.2136/sssaj2000.642681x, 2000.*
>
> *Six, J., Bossuyt, H., Degryze, S., and Denef, K.: A history of research on the link between (micro)aggregates, soil biota, and soil organic matter dynamics, Soil and Tillage Research, 79, 7-31, https://doi.org/10.1016/j.still.2004.03.008, 2004.*
>
> *Loaiza Puerta, V., Pujol Pereira, E. I., Wittwer, R., van der Heijden, M., and Six, J.: Improvement of soil structure through organic crop management, conservation tillage and grass-clover ley, Soil and Tillage Research, 180, 1-9, https://doi.org/10.1016/j.still.2018.02.007, 2018.*

Appendix A already give a good guidance what is of interest. Basically the 4 sites are nearly always significantly different for each parameter, not supprising based on the selecting of the sites with this large climate gradient. Biocrust has no effect on coarser fractions (e.g. sand), C and C/N which could be organic C vs inorganic C reflection between sites. Largest and smallest aggregate size fraction, adn R <2. What maybe wortwile is to look at the site * biocrust interaction what it tells about the aggregate stability /biocrust issue

> *This is a good idea, thank you. We added interaction visualizations in Figure 2 and indicated with the presence of letters display on top of the error bars. Further, interaction results has been additionally described in the lines 263 ff for EC, BD, SIC, $N_t$, C/N and clay.*

---

## Author Response (AR2)

Comments to the author:

Thank you for carefully revising your manuscript. As your manuscript is about biocrusts, which may present different characteristics as compared to soil aggregates taken from mineral soils, I suggest that you remove the first part of your introduction, which deal with soil aggregates in general (l. 1-61).

The conclusions need to be improved. The should be more synthetic instead of just summerizing the resultsThe second sentence coul bechanged to We show that biocrusts play a role in soil aggregate stability along the climate gradient with effects on.....(here all effects should be listed including those of the third paragraph). Thereafter the conclusions should be drawn.

> *Dear Reviewer,*
> *We appreciate the interest and the time spent to improve our manuscript. We agree with the suggested changes, so we have included them in the new version of the manuscript.*
> *Kind regards,*
> *Nicolás Riveras-Muñoz, on behalf of all co-authors*